# Research on China's embodied carbon import and export trade from the perspective of value-added trade

**Guangyao Deng[1], Fengying Lu[1], Xiaofang Yue[2]***

**1** School of Statistics, Lanzhou University of Finance and Economics, Lanzhou, PR China, **2** China Center for Special Economic Zone Research, Shenzhen University, Shenzhen, PR China

* harborangel@163.com

## Abstract

The development of globalization has separated the production and consumption of products spatially, and the international trade of products has become a carrier of embodied carbon trade. This paper adopted the perspective of value-added trade to calculate the amount of embodied carbon trade of China from 2006 to 2015 and perform a structural decomposition analysis of the changes in China's embodied carbon trade. This study found that: (1) China's embodied carbon exports are much larger than its embodied carbon imports, and there are differences between countries. China imported the largest amount of embodied carbon from South Korea, and it exported the largest amount of embodied carbon to the United States. (2) The structural decomposition analysis shows that changes in the value-added carbon emission coefficient during the study period would have caused China's embodied carbon trade to decrease, and changes in value-added trade would have caused China's embodied carbon trade to increase. Therefore, countries trading with China need to strengthen their cooperation with China in energy conservation, emission reduction, and product trade. In order to accurately reflect China's embodied carbon trade, it is necessary to calculate embodied carbon trade from the perspective of value-added trade.

## Introduction

With rapid economic growth, China's consumption of various types of fossil energy has been increasing. Consumption of fossil energy generates a large amount of $CO_2$. Since the beginning of the 21st century, China has surpassed the United States to become the world's largest $CO_2$ emission country [1, 2]. In order to reduce carbon emissions, the Chinese government promised to reduce carbon emission intensity by 40–45% in 2020 and by 60–65% in 2030 compared with 2005 [3]. For curtailing carbon emissions, the focus areas include but are not limited to the carbon emissions generated during the production of goods and the transfer of carbon emissions between countries that engage in trading products and services, that is, embodied carbon trade [4]. This paper takes China as an example to study embodied carbon trade.

**Data Availability Statement:** The data supporting the results of the study can be found at: https://worldmrio.com. The necessary files are titled: Eora26Structure.xlsx, Eora26_2015_bp.zip,

Eora26_2014_bp.zip, Eora26_2013_bp.zip, Eora26_2012_bp.zip, Eora26_2011_bp.zip, Eora26_2010_bp.zip, Eora26_2009_bp.zip, Eora26_2008_bp.zip,Eora26_2007_bp.zip, Eora26_2006_bp.zip. The author confirms that they have no special access to this data set.

**Funding:** This study was supported in part by the Gansu Province Double First-class Scientific Research Key Project in the form of funds to GD [GSSYLXM-06].

**Competing interests:** The authors have declared that no competing interests exist.

Input-output model is a kind of economic mathematical model to comprehensively analyze the quantitative dependence relationship between input-output in economic activities. Researchers usually extend the conventional input-output model to the environmental input-output model to study the embodied carbon trade [5], virtual land [6] and embodied energy [7]. According to the number of regions, input–output models can be divided into single-region input–output models and multi-region input–output models. Some researchers have used single-region input–output models to study China's embodied carbon emissions [8–10] and the embodied carbon emissions of specific provinces [11, 12]. Other papers have used multi-region input–output models to study the embodied carbon trade between China and Japan [13–15]; China and the United States [16]; China and Germany [17]; China and India [18]; and China and multiple countries [19]. The embodied carbon trade between multiple regions in China has also been examined [20–25]. Some studies analyzed the embodied carbon trade between major countries in the world [26–28]. It can be accepted based on the aforementioned literature that the input–output model is widely used to calculate embodied carbon trade. Although the above studies have presented research findings on the accounting of embodied carbon trade from the perspective of conventional trade, they have not considered the accounting of embodied carbon trade from the perspective of value-added trade.

In recent years, production and trade in and among countries (regions) around the world have become more and more closely tied. Predominantly, the production of a product involves multiple companies in multiple countries (regions). Production links and their added value vary across different countries and regions. The resulting trade of value-added products is called value-added trade [29, 30]. The traditional statistical methods of import and export trade based on cross-border and final products can no longer accurately reflect the production process of products in today's global value chain and the value-added characteristics in each production link in different countries. For this reason, some researchers have calculated the carbon emissions embodied in the value-added trade of products; that is, they calculated the embodied carbon trade from the perspective of value-added trade [31, 32]. Xu et al. [31] and Zhang et al. [32] pointed out that the embodied carbon trade volume calculated from the perspective of traditional trade is greater than that calculated from the perspective of value-added trade. However, Xu et al. [31] and Zhang et al. [32] limited their studies to the embodied carbon trade between China and the United States and between China and South Korea and did not explore the embodied carbon trade between China and multiple countries (regions). In this context, structural decomposition analysis (SDA) can be useful to study the causes of changes in China's embodied carbon trade. Except Xu et al. [31] and Zhang et al. [32], some researchers calculated China's carbon emissions and measured China's environmental losses from the perspective of value-added trade [33, 34], some calculated the embodied carbon trade among eight regions in China from the perspective of value-added chain [35], and some analyzed carbon emissions from a global value chain perspective [36].

In addition to studies on embodied carbon trade from the perspective of value-added trade, the relationship between carbon emission and added value from the perspective of embodied carbon intensity was also explored [37, 38]. Su and Ang [37] firstly propose the aggregate embodied intensity (AEI) framework by defining the AEI indicator as the ratio of embodied energy (or emissions) to embodied value added using the I-O framework. Recently, the AEI analysis has been further extended to the transmission layer by Su et al. [38]. The AEI indicator at the higher level can be represented as a weighted sum of the AEI indicators at the lower level. There are already studies using the AEI indicators at the country level, such as China [37, 38] and India [39], and at the global level, such as Yang and Su [40] and Duan and Yan [34] using the WIOD database. These studies also use the SDA technique to investigate the driving forces to the changes observed.

The contributions of this paper are as follows: (1) From the perspective of value-added trade, this paper constructs a multi-region input–output model to calculate the embodied carbon imports and exports between China and major countries (regions) from 2006 to 2015. This study also analyzed the differences in the embodied carbon trade of various industries in China. (2) Using the structural decomposition method, this paper investigates the impact of factors such as value-added carbon emission coefficient and value-added trade on the changes in China's embodied carbon import and export trade from 2006 to 2015. (3) By defining the value-added carbon emission coefficient, the influence of intermediate product trade on carbon emission is eliminated. Furthermore, by combining carbon emissions with value-added trade, it avoids double-counting of cross-border trade, thus making the calculation of carbon emissions embodied in trade of goods and services more scientific and reasonable.

## Methods

### Accounting for embodied carbon trade from the perspective of value-added trade

Traditional trade statistical approach produces a 'statistical illusion' that the calculated trade volume is far greater than the actual trade volume. This is because the main statistical objects are cross-border and final products, and only the last link of these products production is actually carried out in the country and other links are performed abroad. Traditional trade statistics approach treats all of these products as final products entirely produced in the country, and such products will be calculated into the country's import and export trade volume. Therefore, the embodied carbon trade emissions data calculated on the basis of traditional trade statistics may be distorted. Value-added trade is calculated on the basis of the added value of products and services in different economies in the global value chain system, avoiding double counting of cross-border trade, thus making the calculation of embodied carbon trade emissions data more scientific and reasonable. This paper defines the value-added carbon emission coefficient and draws on the value-added trade calculation formula given by Koopman et al. [29] to calculate the embodied carbon emissions from the perspective of value-added trade.

With reference to Koopman et al. [29] the value-added trade (including self-consumption) matrix T (mn × n order) of countries (regions) worldwide is the product of the value-added coefficient matrix V, the Leontief inverse matrix B, and the final use matrix Y:

$$
T = VBY = \begin{pmatrix}
V^1(\sum_{r}^{n} B^{1r} Y^{r1}) & V^1(\sum_{r}^{n} B^{1r} Y^{r2}) & \cdots & V^1(\sum_{r}^{n} B^{1r} Y^{rm}) \\
V^2(\sum_{r}^{n} B^{2r} Y^{r1}) & V^2(\sum_{r}^{n} B^{2r} Y^{r2}) & \cdots & V^2(\sum_{r}^{n} B^{2r} Y^{rm}) \\
\cdots & \cdots & \vdots & \cdots \\
V^n(\sum_{r}^{n} B^{nr} Y^{r1}) & V^n(\sum_{r}^{n} B^{nr} Y^{r2}) & \cdots & V^n(\sum_{r}^{n} B^{nr} Y^{rm})
\end{pmatrix} \tag{1}
$$

where m is the total number of industries, n is the total number of countries (regions), and r is any country (region); the non-diagonal element in the value-added coefficient matrix V is 0, the representative element on the diagonal is $\frac{Va_i^r}{X_i^r}$, that is, the ratio of the added value of the i-th industry of the country (region) r to the total input; the Leontief inverse matrix $B = (I - A)^{-1}$; it is the inverse matrix of the difference between the identity matrix I and the direct consumption

coefficient matrix A. The representative element of the direct consumption coefficient matrix A is the ratio of intermediate input to total input in the input–output table in the EORA database. According to the structure of the table [41, 42], the final use of each country includes household final consumption, non-profit institutions serving households, government final consumption, gross fixed capital formation, changes in inventories, and acquisitions less disposals of valuables. This paper combined the above six items into one in order to obtain the final use matrix Y of mn × n order.

In order to combine value-added trade with embodied carbon trade, this article defines the value-added carbon emission coefficient. It should be noted that it is different from the calculation of embodied carbon trade under the traditional trade perspective, which calculates embodied carbon trade by defining carbon emission intensity coefficient (the ratio of carbon emissions to total input or total output) [15]. This definition does not take into account that the intermediate input part of the product may be imported from abroad, and the carbon emission responsibility caused by the production of imported products from abroad is borne by the foreign country under the producer responsibility system. The value-added carbon emission coefficient is defined as follows:

$$c_i^r = \frac{C_i^r}{Va_i^r} \tag{2}$$

where $C_i^r$ is the carbon emission of the i-th industry in the country (region) r. If the value-added carbon emission coefficient row vector is rewritten into the form of a diagonal matrix (the element on the diagonal is the value-added carbon emission coefficient, and the element on the non-diagonal line is 0), then the embodied carbon export of the country (region) s to the country (region) t is calculated as follows:

$$E^{st} = \hat{c}^s V^s \sum_{r}^{n} B^{sr} Y^{rt} \tag{3}$$

$E^{st}$ is a row vector of order m × 1. $E^{st}$ in Eq (3) can be understood as the embodied carbon imports of country (region) t from country (region) s. The total embodied carbon exports of country (region) s to all other countries (regions) are given by

$$E^s = \sum_{t \neq s}^{n} (\hat{c}^s V^s \sum_{r}^{n} B^{sr} Y^{rt}) \tag{4}$$

Similarly, the total embodied carbon imports of country (region) s from all other countries (regions) are given by

$$F^s = \sum_{t \neq s}^{n} (\hat{c}^t V^t \sum_{r}^{n} B^{tr} Y^{rs}) \tag{5}$$

## Structural decomposition analysis

Structural decomposition analysis is a method to analyze the impact of changes in the components of economic variables on total changes. The embodied carbon imports and exports in this paper include four factors c, V, B, and Y. According to Dietzenbacher and Los [43], there are 4! (24) different decomposition methods for changes in embodied carbon imports and exports. To solve the problem of inconsistent decomposition methods, with reference to Dietzenbacher and Los [38], this paper used the following two-pole decomposition method to separately conduct a structural decomposition analysis of changes in embodied carbon exports

(imports):

$$\Delta E^s = \Delta E_1^s + \Delta E_2^s + \Delta E_3^s + \Delta E_4^s \tag{6}$$

$$\Delta E_1^s = \frac{1}{2}\left(\sum_{t \neq s}^{n} \Delta \hat{c}^s \left(V^s(1)\sum_{r}^{n} B^{sr}(1)Y^{rt}(1) + V^s(0)\sum_{r}^{n} B^{sr}(0)Y^{rt}(0)\right)\right) \tag{7}$$

$$\Delta E_2^s = \frac{1}{2}\left(\sum_{t \neq s}^{n}\hat{c}^s(0)\Delta V^s \sum_{r}^{n}B^{sr}(1)Y^{rt}(1) + \sum_{t \neq s}^{n}\hat{c}^s(1)\Delta V^s \sum_{r}^{n}B^{sr}(0)Y^{rt}(0)\right) \tag{8}$$

$$\Delta E_3^s = \frac{1}{2}\left(\sum_{t \neq s}^{n}\hat{c}^s(0)V^s(0)\sum_{r}^{n}\Delta B^{sr}Y^{rt}(1) + \sum_{t \neq s}^{n}\hat{c}^s(1)V^s(1)\sum_{r}^{n}\Delta B^{sr}Y^{rt}(0)\right) \tag{9}$$

$$\Delta E_4^s = \frac{1}{2}\left(\sum_{t \neq s}^{n}\hat{c}^s(0)V^s(0)\sum_{r}^{n}B^{sr}(0)\Delta Y^{rt} + \sum_{t \neq s}^{n}\hat{c}^s(1)V^s(1)\sum_{r}^{n}B^{sr}(1)\Delta Y^{rt}\right) \tag{10}$$

where $\Delta$ is the increment, 1 is the value at the end of the period (such as the value in 2015), and 0 is the value at the beginning of the period (such as the value in 2006). $\Delta E^s$ is the change in embodied carbon exports of country (region) s, $\Delta E_1^s$ is the change in embodied carbon exports caused by the change in the value-added carbon emission coefficient, $\Delta E_2^s$ is the change in embodied carbon exports caused by the change in the value-added coefficient, $\Delta E_3^s$ is the change in embodied carbon exports caused by the change in the Leontief inverse matrix, and $\Delta E_4^s$ is the change in embodied carbon exports caused by the change in the final use matrix. The total of $\Delta E_2^s$, $\Delta E_3^s$, and $\Delta E_4^s$ is the change in embodied carbon exports caused by the change in value-added trade. Similarly, $\Delta F^s$ can be defined as the change in embodied carbon imports in country (region) s, $\Delta F_1^s$ is the change in embodied carbon imports caused by the change in the value-added carbon emission coefficient, $\Delta F_2^s$ is the change in embodied carbon imports caused by the change in the value-added coefficient, $\Delta F_3^s$ is the change in embodied carbon imports caused by the change in the Leontief inverse matrix, and $\Delta F_4^s$ is the change in embodied carbon imports caused by the change in the final use matrix. The specific equation is as follows:

$$\Delta F^s = \Delta F_1^s + \Delta F_2^s + \Delta F_3^s + \Delta F_4^s \tag{11}$$

$$\Delta F_1^s = \frac{1}{2}\left(\sum_{t \neq s}^{n}\Delta\hat{c}^t\left(V^t(1)\sum_{r}^{n}B^{tr}(1)Y^{rs}(1) + V^t(0)\sum_{r}^{n}B^{tr}(0)Y^{rs}(0)\right)\right) \tag{12}$$

$$\Delta F_2^s = \frac{1}{2}\left(\sum_{t \neq s}^{n}\hat{c}^t(0)\Delta V^t \sum_{r}^{n}B^{tr}(1)Y^{rs}(1) + \sum_{t \neq s}^{n}\hat{c}^t(1)\Delta V^t \sum_{r}^{n}B^{tr}(0)Y^{rs}(0)\right) \tag{13}$$

$$\Delta F_3^s = \frac{1}{2}\left(\sum_{t \neq s}^{n}\hat{c}^t(0)V^t(0)\sum_{r}^{n}\Delta B^{tr}Y^{rs}(1) + \sum_{t \neq s}^{n}\hat{c}^t(1)V^t(1)\sum_{r}^{n}\Delta B^{tr}Y^{rs}(0)\right) \tag{14}$$

$$\Delta F_4^s = \frac{1}{2}\left(\sum_{t \neq s}^{n}\hat{c}^t(0)V^t(0)\sum_{r}^{n}B^{tr}(0)\Delta Y^{rs} + \sum_{t \neq s}^{n}\hat{c}^t(1)V^t(1)\sum_{r}^{n}B^{tr}(1)\Delta Y^{rs}\right) \tag{15}$$

where the total of $\Delta F_2^s$, $\Delta F_3^s$, and $\Delta F_4^s$, is the change in embodied carbon imports caused by the change in value-added trade.

It should be noted that the input-output model mainly has the following three assumptions: (1) homogeneity assumption: Assuming that each product department produces only one

homogeneous product, and the products of one product department cannot be replaced by the other. That is, the consumption structure, production process and economic use are the same, which is also called the aggregate of homogeneous products. (2) The proportionality hypothesis: The output of the department is directly proportional to the input. Only in this way can output and input be guaranteed to be linear functions. (3) Assumption of relative stability of consumption coefficients: it is assumed that in a certain period (1 year), all kinds of consumption coefficients are relatively stable. This paper is based on the above three assumptions. In addition, the data required for compiling the global input-output table are collected from different countries, so the data in the input-output table are the data after integration, and there will be some differences between the integrated data and the original data.

## Data

The world input–output table and carbon emission data used in this paper are from the EORA database [41, 42]. The EORA database provides multi-regional and multi-sectoral input–output tables of 26 industries (The specific names of the 26 industries in the 190 countries (regions) can be found in the EORA database (http://www.worldmrio.com/) in 190 countries (regions) from 1990 to 2015). In order to analyze the changing trend of China's embodied carbon trade in recent years, this paper selected the data from 2006 to 2015 for research (Researchers can also choose other time periods for research, such as 2000–2015). The EORA database also provides carbon emissions accounting data from multiple institutions. This paper used the data from the EDGAR database created by the European Commission and the Netherlands Environmental Assessment Agency. In addition, this paper studied China (In this paper, China refers specifically to Mainland China, with Hong Kong, Macao, and Taiwan listed separately) as an example to describe the calculation results of embodied carbon trade from the perspective of value-added trade and then conducted a structural decomposition analysis of the changes in embodied carbon imports and exports. In order to make the data of different years comparable, this paper uses 2006 as the base period to deflate data related to prices.

## Results and discussions

### Accounting results of embodied carbon trade

Using Eqs (4) and (5), the paper obtained the calculation results of China's embodied carbon imports and exports from the perspective of value-added trade in the period 2006–2015, as shown in Fig 1.

It can be seen from Fig 1: (1) From 2006 to 2015, China's embodied carbon exports were much larger than its embodied carbon imports. This is because China's exports of products and services have invariably far exceeded its imports; that is, China has always had a trade surplus. (2) From 2006 to 2015, China's embodied carbon imports generally showed an increasing trend except falling slightly from 2013 to 2015. Unlike embodied carbon imports, China's embodied carbon exports experienced multiple cycles of change of increase and decrease from 2006 to 2015.

In order to analyze the country-to-country differences in China's embodied carbon trade, this paper took as an example (According to the calculation results, consistent with 2015, China still had the highest percentage of embodied carbon imports from the following 10 countries in 2006–2014: Australia, Germany, India, Indonesia, Japan, Kazakhstan, Malaysia, South Korea, Russia, and USA. Consistent with 2015, China still had the highest percentage of embodied carbon exports to the following 10 countries (regions) from 2009 to 2014: Canada, France, Germany, Hong Kong, India, Italy, Japan, South Korea, UK, and USA. However, from

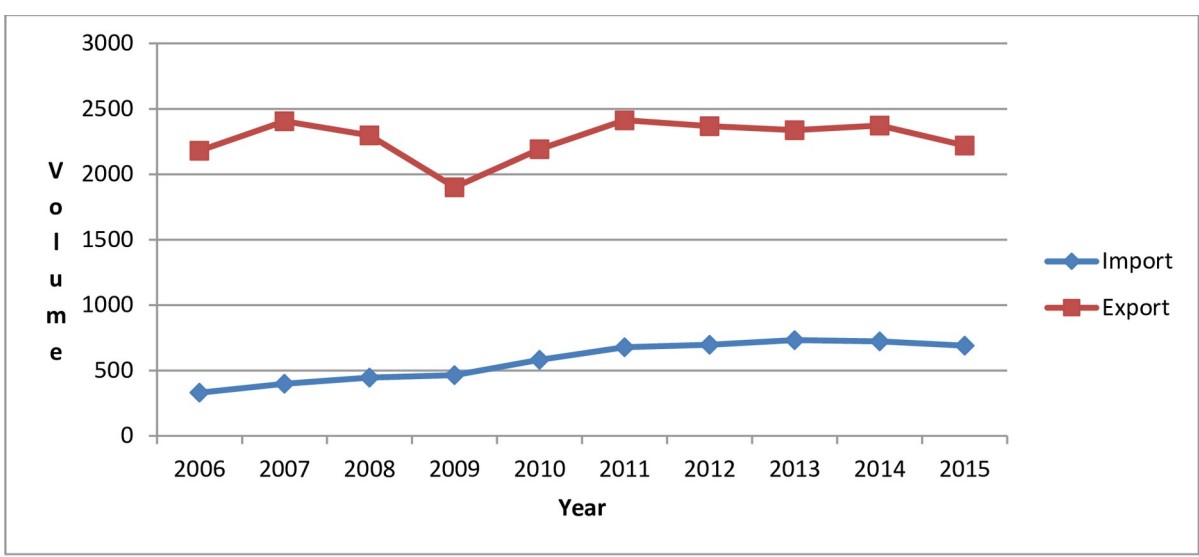

**Fig 1. China's embodied carbon imports and exports from 2006 to 2015 (unit: 10⁶ t).**

2006 to 2008, India was replaced by Spain and dropped to the 11ᵗʰ place) the top 10 countries (regions) that are China's trading partners and with whom China had embodied carbon imports and exports in 2015 to illustrate the changing trends of the embodied carbon imports and exports between China and its major trading partners, as shown in Figs 2 and 3 (ROW1 in Fig 2 refers to other countries (regions) in the world except China, Australia, Germany, India, Indonesia, Japan, Kazakhstan, Malaysia, South Korea, Russia, and USA. ROW2 in Fig 3 refers to other countries (regions) in the world except China, Canada, France, Germany, Hong Kong (China), India, Italy, Japan, South Korea, UK, and USA).

From Figs 2 and 3, we can see: (1) From 2006 to 2015, China's embodied carbon imports from its major trading partners accounted for about 60%, and the embodied carbon exports to its major trading partners accounted for more than 65%. This shows that China's embodied

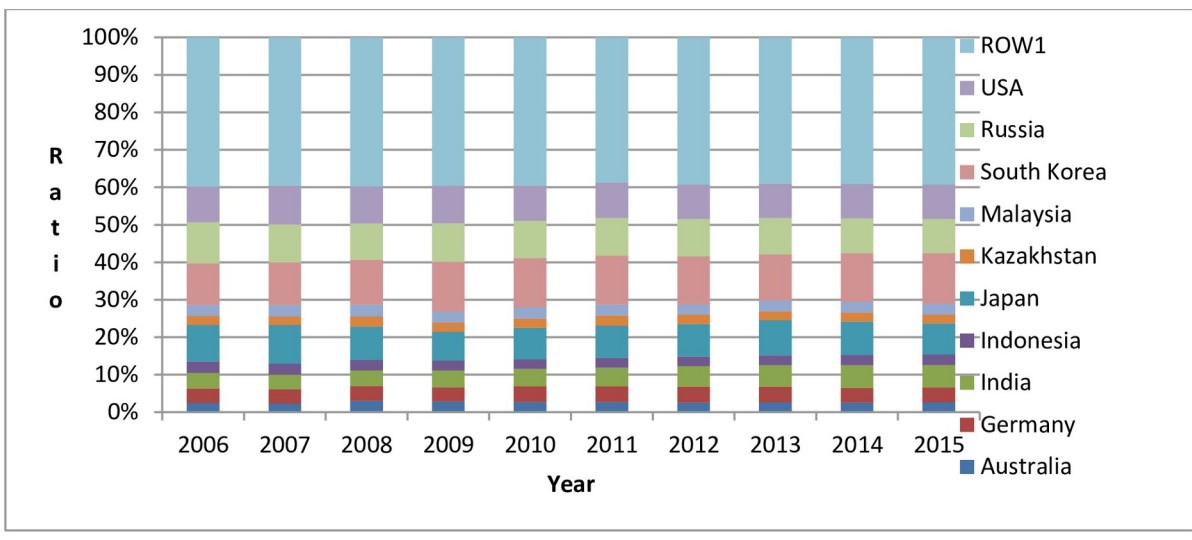

**Fig 2. Percentage of embodied carbon imported by China from major trading partners from 2006 to 2015.**

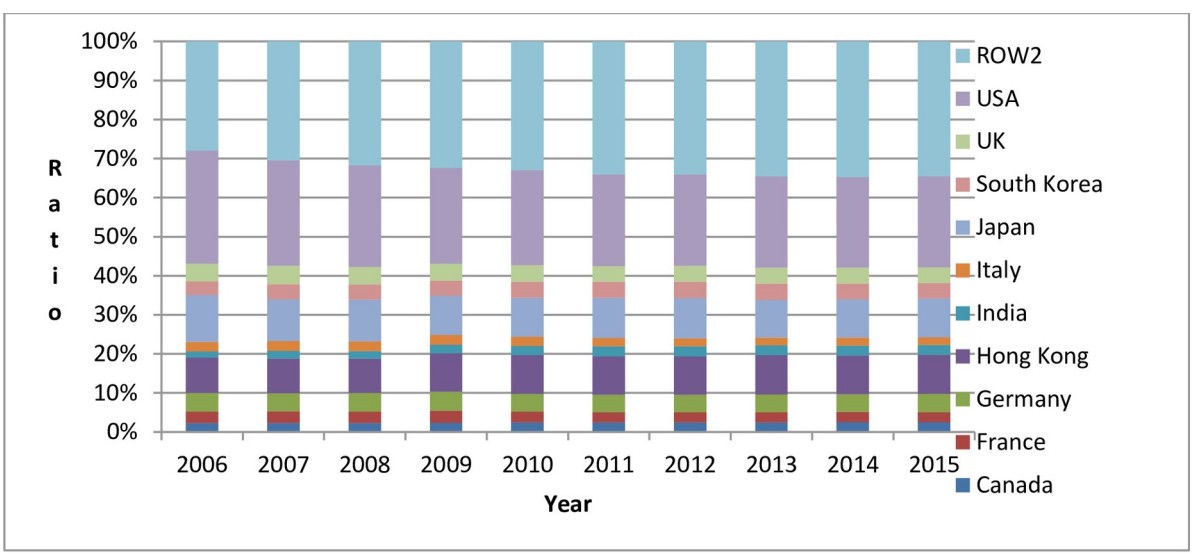

**Fig 3. Percentage of embodied carbon in China's exports to major trading partners from 2006 to 2015.**

carbon exports are more concentrated to a few countries (regions) than are its embodied carbon imports. (2) Among China's major trading partners with respect to embodied carbon imports from 2006 to 2015, China imported the largest amount of embodied carbon from South Korea (From 2006 to 2015, among China's embodied carbon import trading partners, China's annual embodied carbon imports from South Korea accounted for 11.15%, 11.33%, 11.97%, 13.23%, 13.11%, 13.08%, 12.73%, 12.23%, 12.93%, and 13.39% of China's total embodied carbon import). Among China's major trading partners with respect to embodied carbon imports from 2006 to 2015, China exported the largest amount of embodied carbon to the United States (From 2006 to 2015, among China's embodied carbon export trading partners, China's annual embodied carbon exports to the USA accounted for 28.93%, 27.08%, 26.05%, 24.52%, 24.30%, 23.52%, 23.48%, 23.40%, 23.19%, and 23.36% of China's total annual

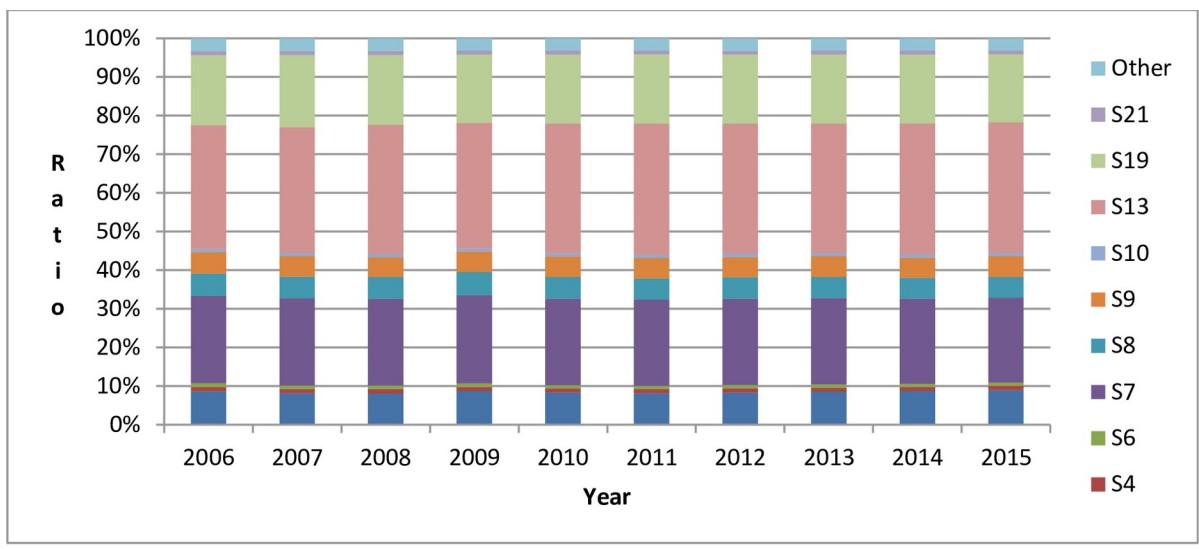

**Fig 4. Percentage of embodied carbon imports in China's major industries from 2006 to 2015.**

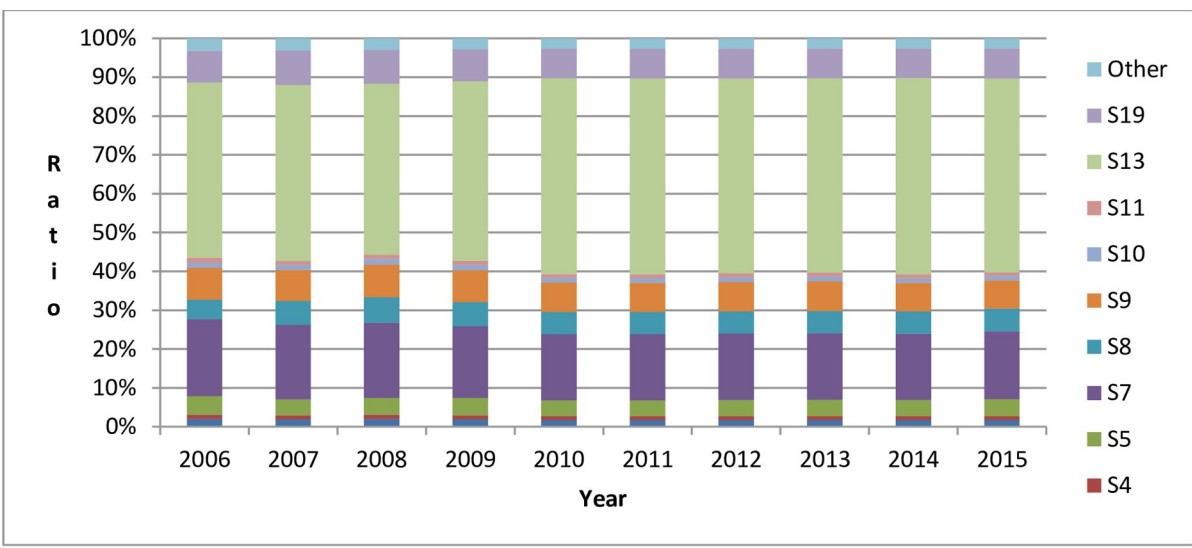

**Fig 5. Percentage of embodied carbon exports in China's major industries from 2006 to 2015.** Note: (1) The order of the industries is the order of the industries in the world input–output table. "Other industries" refers to the 16 industries without the 10 major industries. For specific industry names, please refer to the EORA database. (2) The specific names of various industries in Figs 4 and 5 are S3 (Mining and Quarrying), S4 (Food & Beverages), S5 (Textiles and Wearing Apparel), S6 (Wood and Paper), S7 (Petroleum, Chemical and Non-Metallic Mineral Products), S8 (Metal Products), S9 (Electrical and Machinery), S10 (Transport Equipment), S11 (Other Manufacturing), S13 (Electricity, Gas and Water), S19 (Transport), and S21 (Financial Intermediation and Business Activities).

embodied carbon exports). (3) The comparison of China's major trading partners of embodied carbon imports and exports shows that during the period 2006–2015, China had a large bilateral trade volume with Germany, India, Japan, South Korea, and USA.

In order to analyze inter-industry differences in China's embodied carbon trade, this paper used China's industries whose percentage of embodied carbon imports and exports ranked top 10 in 2015 as examples to illustrate the changing trends of embodied carbon imports and exports of China's major industries from 2006 to 2015, as shown in Figs 4 and 5.

From Figs 4 and 5, we can see: (1) Whether it is embodied carbon imports or exports, the sum of the percentages of China's 10 major industries from 2006 to 2015 are all over 90%. Among them, the industries with the highest percentage of embodied carbon imports and exports are the ones in the S13 category (electricity, gas, and water). In these industries, from 2006 to 2015, the percentages of the embodied carbon imports were all above 30%, and the percentages of the embodied carbon exports were all above 40%. (2) The following industries appear in both Figs 4 and 5: S4 (Food & Beverages), S7 (Petroleum, Chemical and Non-Metallic Mineral Products), S8 (Metal Products), S9 (Electrical and Machinery), S10 (Transport Equipment), S13 (Electricity, Gas and Water), and S19 (Transport). In the global value chain, different stages of a product's production (e.g., design, production, assembly, marketing, and after-sales service) are carried out in multiple countries, which results in a large volume of imports and exports of products in these industries, so industries with more embodied carbon imports may also have more embodied carbon exports.

## Structural decomposition analysis

This paper analyzed the structural decomposition of the changes in China's embodied carbon imports and exports from 2006 to 2015. The structure decomposition analysis results of the changes in the total embodied carbon imports (exports) are shown in Table 1 (The structural decomposition analysis that changes year by year (for example, 2006–2007) is the same as the

**Table 1. Structural decomposition of changes in the trade volume of China's embodied carbon imports and exports from 2006 to 2015 (unit: $10^6$ t).**

| Imports/exports | Change in trade volume | First item | Second item | Third item | Fourth item | Sum of the last three items |
|---|---|---|---|---|---|---|
| Imports | 360.1010 | −1211.2790 | 937.6584 | −50.5470 | 684.2685 | 1571.3799 |
| Exports | 40.1613 | −1837.2290 | −111.5835 | 835.0070 | 1153.9668 | 1877.3903 |

structural decomposition analysis for the overall time period (2006–2015). Therefore, this paper does not demonstrate the former.).

It can be seen from Table 1: (1) Compared with embodied carbon exports, China's embodied carbon imports changed more in the period 2006–2015, which is consistent with the results in Fig 1. (2) For each decomposition item, whether it is embodied carbon imports or exports, the first item is less than 0, and the fourth item is greater than 0. For most industries in most countries, compared with 2006, energy-saving and emission reduction technologies were improving in 2015, and the carbon emissions generated by the production of value-added products per unit decreased. That is, the value-added carbon emission coefficient c decreased. Changes in the value-added carbon emission coefficient reduced embodied carbon trade; that is, the first item is negative. Compared with 2006, China's imports of products from most countries (regions) increased in 2015, and its exports to most countries (regions) also increased. This led to an increase in most of the elements in the final use matrix $Y$, which led to an increase in China's embodied carbon imports and exports—that is, the fourth item is positive. In addition, there were insignificant changes in the ratio of added value to total input and the ratio of intermediate input to total input in various industries of different countries during the study period. In other words, the changes in the value-added coefficient and the Leontief inverse matrix were not obvious, but generally, the changes in the coefficient $V$ of added value led to an increase in China's embodied carbon imports and a decrease in its embodied carbon exports, and the changes in the Leontief inverse matrix $B$ led to a decrease in China's embodied carbon imports and an increase in its embodied carbon exports. This is because the change in the coefficient $V$ of added value and the change in the Leontief inverse matrix $B$ can be positive or negative. (3) From the sum of the second, third, and fourth terms, whether they are imports or exports, compared with 2006, the increase in value-added trade in 2015 led to an increase in China's embodied carbon trade.

Next, this paper considered Australia, Germany, India, Indonesia, Japan, Kazakhstan, Malaysia, South Korea, Russia, USA, and ROW1 in Fig 2 as examples to illustrate the differences due to the countries (regions) in the structural decomposition analysis of the changes in China's embodied carbon imports from 2006 to 2015, as shown in Table 2.

It can be seen from Table 2: (1) Compared with 2006, China's embodied carbon imports from major import sources increased in 2015. Except for ROW1, China's embodied carbon imports from South Korea increased the most (by $55.6320 \times 10^6$ t). (2) The change in the value-added carbon emission coefficient from 2006 to 2015 led to a reduction in China's embodied carbon imports; that is, $\Delta F_1$ was less than zero. Except for ROW1, among China's sources of imports, Russia had the largest absolute value of this decomposition item. The change in the value-added carbon emission coefficient caused China's embodied carbon import from Russia to drop by $62.6757 \times 10^6$ t. (3) The changes in the coefficient of added value from 2006 to 2015 led to an increase in China's embodied carbon imports from Australia, India, Indonesia, Kazakhstan, Malaysia, Russia, and the United States, but the embodied carbon imports from Germany, Japan, and South Korea decreased. (4) The changes in the Leontief inverse matrix from 2006 to 2015 led to a reduction in China's embodied carbon imports from Australia, Germany, India, Japan, Kazakhstan, Malaysia, and the United States, but the embodied carbon imports from Indonesia, South Korea, and Russia increased. (5) The

**Table 2. Structural decomposition analysis of changes in China's embodied carbon imports from 2006 to 2015 (breakdown by import source) (unit: $10^6$ t).**

| Country/region | $\Delta F$ | $\Delta F_1$ | $\Delta F_2$ | $\Delta F_3$ | $\Delta F_4$ | Sum of the last three items |
|---|---|---|---|---|---|---|
| Australia | 9.8022 | −7.5126 | 0.2264 | −1.2406 | 18.3289 | 17.3147 |
| Germany | 14.6621 | −7.0504 | −4.0893 | −1.7262 | 27.5280 | 21.7125 |
| India | 27.8140 | −6.7307 | 4.1085 | −3.2980 | 33.7341 | 34.5446 |
| Indonesia | 9.5571 | −12.2381 | 1.1605 | 0.0244 | 20.6103 | 21.7953 |
| Japan | 24.7192 | −11.5012 | −2.1816 | −23.4014 | 61.8034 | 36.2205 |
| Kazakhstan | 8.0590 | −11.0611 | 3.1305 | −0.8705 | 16.8600 | 19.1200 |
| Malaysia | 11.1953 | −7.1564 | 2.1551 | −4.1308 | 20.3273 | 18.3516 |
| South Korea | 55.6320 | −27.6151 | −5.8583 | 7.8246 | 81.2808 | 83.2471 |
| Russia | 27.1094 | −62.6757 | 13.5146 | 5.9624 | 70.3082 | 89.7852 |
| USA | 31.9679 | −16.0375 | 5.0554 | −19.8658 | 62.8158 | 48.0054 |
| ROW1 | 139.5827 | −1041.7003 | 920.4366 | −9.8252 | 270.6716 | 1181.2830 |
| Total | 360.1010 | −1211.2790 | 937.6584 | −50.5470 | 684.2685 | 1571.3799 |

changes in the final use matrix from 2006 to 2015 led to an increase in China's embodied carbon imports. Except for ROW1, China's embodied carbon imports from South Korea increased the most (by $81.2808 \times 10^6$ t). (6) As indicated by the sum of the last three items in the decomposition items, the changes in China's value-added imports from various countries (regions) from 2006 to 2015 all led to an increase in China's embodied carbon imports. Except for ROW1, Russia had the largest corresponding value. The change in value-added imports caused China's embodied carbon imports from Russia to increase by $89.7852 \times 10^6$ t.

This paper also considered Canada, France, Germany, Hong Kong, India, Italy, Japan, South Korea, UK, USA, and ROW2 in Fig 3 as examples to illustrate the differences due to the countries (regions) in the structural decomposition analysis of the changes in China's embodied carbon exports from 2006 to 2015, as shown in Table 3.

It can be seen from Table 3: (1) Compared with 2006, China's embodied carbon exports to France, Italy, Japan, the United Kingdom, and the United States decreased in 2015. Based on the total value of the last three items, China's value-added trade with these countries increased. Together with Eq (4), it can be seen that the reason for the decrease of China's embodied carbon exports to these countries is the decrease in the value-added carbon emission coefficient. (2) Except for ROW2, in China's export destinations, the changes in the value-added carbon

**Table 3. Structural decomposition analysis of changes in China's embodied carbon exports from 2006 to 2015 (breakdown by export destination) (unit: $10^6$ t).**

| Country/region | $\Delta E$ | $\Delta E_1$ | $\Delta E_2$ | $\Delta E_3$ | $\Delta E_4$ | Sum of the last three items |
|---|---|---|---|---|---|---|
| Canada | 7.2058 | −44.8910 | −3.4588 | 23.6875 | 31.8682 | 52.0968 |
| France | −8.4487 | −48.9342 | −3.0029 | 20.4698 | 23.0186 | 40.4855 |
| Germany | 0.8556 | −86.6382 | −5.6110 | 44.7931 | 48.3117 | 87.4938 |
| Hong Kong | 23.9487 | −176.9654 | −10.6867 | 60.2149 | 151.3860 | 200.9141 |
| India | 22.0039 | −40.8972 | −2.7968 | 24.5449 | 41.1530 | 62.9011 |
| Italy | −9.5057 | −38.8024 | −2.4843 | 18.4472 | 13.3338 | 29.2967 |
| Japan | −40.2495 | −196.1645 | −10.1246 | 77.1129 | 88.9269 | 155.9151 |
| South Korea | 10.3527 | −68.9072 | −4.1642 | 32.1122 | 51.3120 | 79.2599 |
| UK | −8.6905 | −77.5783 | −4.3335 | 39.2607 | 33.9606 | 68.8878 |
| USA | −112.0153 | −463.7046 | −25.8595 | 210.2456 | 167.3032 | 351.6894 |
| ROW2 | 154.7042 | −593.7459 | −39.0611 | 284.1183 | 503.3929 | 748.4501 |
| Total | 40.1613 | −1837.2290 | −111.5835 | 835.0070 | 1153.9668 | 1877.3903 |

**Table 4. Industry differences in structural decomposition analysis of China's embodied carbon imports from 2006 to 2015 (unit: $10^6$ t).**

| Industry | $\Delta F$ | $\Delta F_1$ | $\Delta F_2$ | $\Delta F_3$ | $\Delta F_4$ | Sum of the last three items |
|---|---|---|---|---|---|---|
| S3 | 33.3749 | −238.0949 | 210.2026 | 0.9457 | 60.3215 | 271.4698 |
| S4 | 3.8337 | −4.9434 | 2.3897 | −0.5428 | 6.9303 | 8.7772 |
| S6 | 2.6402 | −3.4085 | 0.8697 | −1.3691 | 6.5482 | 6.0488 |
| S7 | 76.6707 | −73.6984 | 13.5865 | −14.3769 | 151.1595 | 150.3691 |
| S8 | 18.4550 | −18.3871 | 2.9654 | −5.8061 | 39.6828 | 36.8421 |
| S9 | 18.7686 | −14.7665 | 2.7165 | −8.1106 | 38.9291 | 33.5350 |
| S10 | 2.9594 | −7.6257 | 6.0119 | −1.0917 | 5.6649 | 10.5851 |
| S13 | 127.6438 | −713.1131 | 617.5038 | −1.3681 | 224.6212 | 840.7569 |
| S19 | 61.6144 | −70.6272 | 30.9567 | −21.0867 | 122.3715 | 132.2415 |
| S21 | 3.3573 | −6.9402 | 3.7205 | −0.2594 | 6.8364 | 10.2975 |
| Other | 10.7830 | −59.6739 | 46.7352 | 2.5187 | 21.2030 | 70.4568 |
| Total | 360.1010 | −1211.2790 | 937.6584 | −50.5470 | 684.2685 | 1571.3799 |

emission coefficient, the value-added coefficient, the Leontief inverse matrix, the final use matrix, and value-added trade all had great impacts on China's embodied carbon export to the United States, amounting to $-463.7046 \times 10^6$ t, $-25.8595 \times 10^6$ t, $210.2456 \times 10^6$ t, $167.3032 \times 10^6$ t, and $351.6894 \times 10^6$ t, respectively. (3) In each of the decomposition items, the changes in the value-added carbon emission coefficient and the value-added coefficient had a negative impact on China's embodied carbon exports. The changes in the Leontief inverse matrix, the final use matrix, and the value-added trade were all positive.

Next, this paper considered the industries S3 (Mining and Quarrying), S4 (Food & Beverages), S6 (Wood and Paper), S7 (Petroleum, Chemical and Non-Metallic Mineral Products), S8 (Metal Products), S9 (Electrical and Machinery), S10 (Transport Equipment), S13 (Electricity, Gas and Water), S19 (Transport), and S21 (Financial Intermediation and Business Activities) in Fig 4 as examples to illustrate the differences due to the countries (regions) in the structural decomposition analysis of the changes in China's embodied carbon imports from 2006 to 2015, as shown in Table 4.

It can be seen from Table 4: (1) For each major industry, China's embodied carbon imports increased from 2006 to 2015; that is, $\Delta F$ was greater than zero. Although changes in the value-added carbon emission coefficient would have reduced China's embodied carbon imports ($\Delta F_1$ less than 0), the increase in value-added trade led to an increase in China's embodied carbon imports (the sum of the last three items is greater than zero). Overall, compared with 2006, China's embodied carbon imports increased by $360.1010 \times 10^6$ t in 2015. (2) Among all industries, the S13 industries (electricity, gas, and water) had the largest change in embodied carbon imports, which was $127.6438 \times 10^6$ t. The absolute values of the first, second, and fourth items and the absolute value of the sum of the last three items of the structural decomposition analysis in these industries were the largest, which were $-713.1131 \times 10^6$ t, $617.5038 \times 10^6$ t, $224.6212 \times 10^6$ t, and $840.7569 \times 10^6$ t, respectively. (3) The changes in the value-added coefficient from 2006 to 2015 would have increased the embodied carbon imports in various industries in China, but the changes in the Leontief inverse matrix led to an increase in the embodied carbon import in some industries, such as mining and quarrying in the S3 category, and a decrease in embodied carbon imports in other industries, namely S4 (Food & Beverages), S6 (Wood and Paper), S7 (Petroleum, Chemical and Non-Metallic Mineral Products), S8 (Metal Products), S9 (Electrical and Machinery), S10 (Transport Equipment), S13 (Electricity, Gas and Water), S19 (Transport), and S21 (Financial Intermediation and Business Activities).

**Table 5. Industry differences in structural decomposition analysis of changes in China's embodied carbon exports from 2006 to 2015 (unit: $10^6$ t).**

| Industry | $\Delta E$ | $\Delta E_1$ | $\Delta E_2$ | $\Delta E_3$ | $\Delta E_4$ | Sum of the last three items |
|---|---|---|---|---|---|---|
| S3 | −3.7891 | −27.5482 | −3.8010 | 7.6762 | 19.8839 | 23.7591 |
| S4 | −3.3709 | −23.3201 | 3.0322 | 5.3988 | 11.5182 | 19.9492 |
| S5 | −8.4074 | −106.7736 | 17.6450 | 25.2719 | 55.4493 | 98.3663 |
| S7 | −43.8769 | −463.2943 | 66.6552 | 143.0875 | 209.6746 | 419.4174 |
| S8 | 21.0701 | −85.7847 | −29.2330 | 75.3212 | 60.7666 | 106.8548 |
| S9 | −17.2094 | −197.7231 | 29.8135 | 52.3047 | 98.3955 | 180.5137 |
| S10 | −3.0976 | −29.2875 | 3.1632 | 6.6972 | 16.3295 | 26.1898 |
| S11 | −6.6927 | −34.8805 | 10.8258 | 4.1521 | 13.2099 | 28.1878 |
| S13 | 123.3236 | −593.4082 | −267.6612 | 448.1765 | 536.2165 | 716.7318 |
| S19 | −5.1104 | −213.2129 | 52.6447 | 57.2527 | 98.2051 | 208.1025 |
| Other | −12.6779 | −61.9960 | 5.3321 | 9.6682 | 34.3178 | 49.3180 |
| Total | 40.1613 | −1837.2290 | −111.5835 | 835.0070 | 1153.9668 | 1877.3903 |

This paper next used the industries S3 (Mining and Quarrying), S4 (Food & Beverages), S5 (Textiles and Wearing Apparel), S7 (Petroleum, Chemical and Non-Metallic Mineral Products), S8 (Metal Products), S9 (Electrical and Machinery), S10 (Transport Equipment), S11 (Other Manufacturing), S13 (Electricity, Gas and Water), and S19 (Transport) in Fig 5 as examples to illustrate the industry differences in the structural decomposition analysis of China's embodied carbon exports from 2006 to 2015, as shown in Table 5.

It can be seen from Table 5: (1) Compared with 2006, in 2015, the industries in the S8 category (metal products) and S13 category (electricity, gas and water) witnessed an increase in embodied carbon exports, and the embodied carbon exports in the S3 industries (mining and quarrying) decreased. Although changes in the value-added carbon emission coefficient led to a decrease in China's embodied carbon exports (that is, $\Delta E_1$ less than zero), changes in value-added trade still led to an increase in China's embodied carbon exports (the sum of the last three items is greater than zero). Overall, compared with 2006, China's embodied carbon exports increased by $40.1613 \times 10^6$ t in 2015. (2) Among the various industries, the S13 industries (electricity, gas, and water) saw the largest change in embodied carbon exports, and the absolute value of each decomposition item was also the largest, corresponding to $123.3236 \times 10^6$ t, $−593.4082 \times 10^6$ t, $−267.6612 \times 10^6$ t, $448.1765 \times 10^6$ t, $536.2165 \times 10^6$ t, and $716.7318 \times 10^6$ t. (3) The change in the value-added coefficient led to a decrease in embodied carbon exports in industries such as S3 (mining and quarrying), S8 (metal products), and S13 (electricity, gas and water) but an increase in embodied carbon imports in the industries S4 (Food & Beverages), S5 (Textiles and Wearing Apparel), S7 (Petroleum, Chemical and Non-Metallic Mineral Products), S9 (Electrical and Machinery), S10 (Transport Equipment), S11 (Other Manufacturing), S13 (Electricity, Gas and Water), and S19 (Transport). The changes in the Leontief inverse matrix also led to an increase in embodied carbon exports in various industries, as can be observed in Table 5.

The comparison of the results in Tables 2 and 3 (or the results in Tables 4 and 5) shows that the first decomposition item is negative, and the fourth decomposition item and the sum of the final three items are positive. This shows that the changes in the value-added carbon emission coefficient during the period 2006–2015 reduced China's embodied carbon trade, but the changes in the final use matrix and value-added trade increased China's embodied carbon trade.

It should be noted that the structural decomposition analysis in this paper only analyzes the changes in China's embodied carbon trade in 2006 and 2015, and does not analyze the changes

in the interim years (such as 2006–2007, 2007–2008, 2008–2009, 2009–2010, 2010–2011, 2011–2012, 2012–2013, 2013–2014, 2014–2015) which means that the changes in the interim years are ignored. In addition, there is neither analysis of the structural changes of the bilateral trade between the two countries (such as the embodied carbon trade between China and the United States) from the industry level; nor further analysis of any special industries (such asS1 (Agriculture)) from a national perspective. If there is any analysis from the above perspectives, more results will come up and the reasons for the changes in China's embodied carbon trade can be explained in depth.

## Discussions

Different from the literature [8–28] that calculates the embodied carbon trade from the perspective of traditional trade, this paper calculates the carbon emissions embodied in the international trade of products in various industries in China from the perspective of value-added trade. The embodied carbon trade calculated from the perspective of traditional trade is usually larger than that calculated from the perspective of value-added trade, especially for industries with a large proportion of intermediate input, because the embodied carbon of intermediate product trade is not stripped out.

Structural decomposition analysis is applied to exploring the impact of changes in value-added trade on embodied carbon trade. This is different from the existing literature [31–36] which either does not adopt the structural decomposition analysis method, or only analyzes the structure of the underlying carbon trade from the source of the global value chain.

In addition, the paper is different from the literature [37, 38] related on the Aggregate intensity (AEI), which mainly discusses the structural decomposition of embodied carbon or embodied energy intensity, while the paper deals with the structural decomposition of embodied carbon trade volume.

The sharp decline in China's embodied carbon exports from 2008 to 2009 was attributed to the subprime mortgage crisis that broke out in 2008. The real economies of various countries were negatively affected in 2009, which led to the shrinking of foreign markets, the decline in China's exports of products and services, and the decline in its embodied carbon exports. The decline in China's embodied carbon imports and exports in 2015 was attributed to the sluggish external demand (in the case of exports) and the sharp drop in international commodity prices (for imports).

The industries with the highest percentage of embodied carbon imports and exports are the ones in the S13 category (electricity, gas, and water). In these industries, from 2006 to 2015, the percentages of the embodied carbon imports were all above 30%, and the percentages of the embodied carbon exports were all above 40%. This is because the production and supply of electricity, gas, and water consume a lot of energy, which generates a lot of carbon emissions (According to the data in the EORA database, the carbon emissions of this industry from 2006 to 2015 accounted for 47%-54% of the total carbon emissions of all industries in China), leading to the highest amount of embodied carbon imports and exports in these industries. It should be noted that the products produced by the production and supply of electricity, gas, and water are mainly used for domestic consumption. China's exports and imports in these industries are relatively small, but owing to the large direct carbon emission coefficient, their embodied carbon imports and exports are still the highest.

## Conclusions and implications

This paper used the world input–output table and carbon emission data in the EORA database to calculate China's embodied carbon trade volume from the perspective of value-added trade

and conducted a structural decomposition analysis of the changes in China's embodied carbon trade. The following are the notable research results: (1) From 2006 to 2015, China's embodied carbon exports were much larger than its embodied carbon imports. China's embodied carbon imports generally showed an increasing trend, and its embodied carbon exports underwent many cycles of changes of increase and decrease. (2) There were country-specific differences in China's embodied carbon trade. From 2006 to 2015, China imported the largest amount of embodied carbon from South Korea, and China exported the largest amount of embodied carbon to the United States. (3) There were industry differences in China's embodied carbon trade. The industries with the highest percentage of embodied carbon imports and exports were all in the S13 category (electricity, gas, and water). From 2006 to 2015, the embodied carbon imports of the S13 category accounted for more than 30% of the total, and its embodied carbon exports accounted for more than 40% of the total. (4) The structural decomposition analysis shows that changes in the value-added carbon emission coefficient from 2006 to 2015 would have caused a decrease in China's embodied carbon trade, and changes in value-added trade would have caused an increase in China's embodied carbon trade. The size of each decomposition item differed by country and industry. Among the major trading partners, China's embodied carbon imports from Russia and its embodied carbon exports to the United States were affected the most by the changes in the value-added carbon emission coefficient and value-added trade. Among the major industries, the embodied carbon imports and exports in the S13 industries (electricity, gas, and water) were affected the most by the changes in the value-added carbon emission coefficient and value-added trade.

Following the above research results, the following policy recommendations can be proposed: (1) Value-added trade avoids double counting of cross-border trade and can more accurately reflect the trade interests of both parties. Therefore, calculating embodied carbon trade from the perspective of value-added trade can help accurately divide the carbon emission responsibilities of countries around the world. As China's embodied carbon exports are much larger than its embodied carbon imports, the traditional emission reduction model of "whoever produces it is held responsible for it" is not suitable for China. This will give rise to the problem of carbon leakage, which will make it difficult to achieve the goal of global $CO_2$ emission reduction. Therefore, the traditional carbon emission reduction model needs to be changed. Countries consuming China's products need to give China some economic compensation. That is, product producers and consumers need to share the responsibility for carbon emission reductions. (2) Because of the country-to-country differences in China's embodied carbon trade, for countries that trade with China more, such as the United States and South Korea, China needs to strengthen its cooperation with them in energy conservation, emission reduction, and product trade. (3) As there were industry differences in China's embodied carbon trade, for industries with high carbon emissions, more attention should be paid to improving their energy conservation and emission reduction efficiency, and necessary restrictions should be placed on the export of high energy consumption and high emission products (e.g., reducing export tax rebates or restricting exports). At the same time, financial support should be increased for industrial upgrading and industrial restructuring to reduce carbon emissions. (4) The increase in value-added trade is the main reason for the increase in embodied carbon trade. The calculation of embodied carbon trade from the perspective of value-added trade can avoid double counting of cross-border trade and more accurately reflect China's embodied carbon trade. This enables China to better control its carbon emissions by reducing exports of high-emission industries and increasing imports of products from high-emission industries.

Due to the availability of data, the research deadline of this paper is only 2015. In recent years, especially since the outbreak of COVID-19, China's foreign trade situation may have

undergone major changes, and thus China's embodied carbon trade will also undergo major changes. Due to the impact of the epidemic, China's foreign trade declined sharply from January 2020 to May 2020. As the Chinese government adopted a series of extraordinary policies to stabilize foreign trade, China's foreign trade gradually recovered from June 2020. Overall, the foreign trade in 2020 still increased slightly compared with that in 2019. In this way, the amount of embodied carbon trade may increase slightly, and the main destination of embodied carbon export and the main source of embodied carbon import may change, so the amount of embodied carbon trade in various industries will also be different from the results in 2015.

The further research direction of this paper is to use the complex network analysis method to study the network characteristics of the embodied carbon trade among major countries around the world, and use the regression model to study the influencing factors of the embodied carbon trade.

## Supporting information

**S1 Appendix. Sector list.**
(DOCX)

**S2 Appendix. Countries (regions) list.**
(DOCX)

## Author Contributions

**Data curation:** Fengying Lu.

**Funding acquisition:** Guangyao Deng.

**Supervision:** Xiaofang Yue.

**Writing – original draft:** Guangyao Deng.

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
