## [Decision Letter · Decision Letter 0]

26 Jul 2021

PONE-D-21-16951

Research on China’s Embodied Carbon Import and Export Trade from the Perspective of Value-Added Trade

PLOS ONE

Dear Dr. Yue,

Thank you for submitting your manuscript to PLOS ONE. After careful consideration, we feel that it has merit but does not fully meet PLOS ONE’s publication criteria as it currently stands. Therefore, we invite you to submit a revised version of the manuscript that addresses the points raised by the editor and reviewers during the review process.

We look forward to receiving your revised manuscript.

Kind regards,

Taoyuan Wei

Academic Editor

PLOS ONE

Journal Requirements:

This work was supported by the Natural Science Foundation of China under Grant [number 71704070]; Natual Social Science Fund(17BJY061); Outstanding Youth Fund of Gansu Province [number 20JR5RA206]; Gansu Provincial Higher Education Research Project [number 2020A-058]; and Program of Lanzhou University of Finance and Economics under Grant [number Lzufe2018B-06].

Additional Editor Comments:

In addition to comments from the reviewers, I have several comments for authors to consider. 1) it would enhance the paper by discussing and comparing their results with other relevant studies on this topic together/after presenting their results. 2) What are the limitations and potential problems of the study, such as strict assumptions of the input-output methods, data quality, and the role of price changes over time? 3) Any potential directions for future research? 4) As the analysis is based on data until 2015, will it still be valid or to what extent will it be valid for the recent years, particularly after the COVID-2019 pandemic? 5) English needs to be improved.

Reviewers' comments:

Reviewer's Responses to Questions

**Comments to the Author**

1. Is the manuscript technically sound, and do the data support the conclusions?

Reviewer #1: Partly

Reviewer #2: Yes

2. Has the statistical analysis been performed appropriately and rigorously? 

Reviewer #1: Yes

Reviewer #2: Yes

3. Have the authors made all data underlying the findings in their manuscript fully available?

Reviewer #1: Yes

Reviewer #2: Yes

4. Is the manuscript presented in an intelligible fashion and written in standard English?

Reviewer #1: No

Reviewer #2: Yes

5. Review Comments to the Author

Reviewer #1: Exploring trade from the value-added perspective is a very important topic worth digging into. The starting point of this paper is new and insightful. The authors have done solid work by using input-output analysis and structural decomposition analysis. Some suggestions on polishing the paper into better shape are as follows:

The focus of the paper, especially the result section, should be put on the importance of incorporating value-added perspective and the difference it made on the results, i.e., how does traditional statistics distort facts and over-estimate trade imbalance and how does incorporating value-added perspective change traditional accounting. Results Section One (Accounting results of embodied carbon trade) largely deals with conclusions already extensively presented. This section is suggested to be shortened.

The writing of the introduction section is subpar compared with the rest of the paper. A detailed list of previous studies on embodied carbon emissions using input-output analysis is redundant. The introduction of the input-output analysis seems a bit unprofessional. For a better introduction, the authors are suggested to refer to the following literature.

Wu X D , Guo J L , Han M Y , et al. An overview of arable land use for the world economy: From source to sink via the global supply chain[J]. Land Use Policy, 2018, 76:201-214.

Chen G Q , Wu X D , Guo J L , et al. Global overview for energy use of the world economy: Household-consumption-based accounting based on the world input-output database (WIOD)[J]. Energy Economics, 2019, 81.

Some minor comments:

This sentence in the abstract is too colloquial: “countries that have a lot of trade with China”.

The first sentence in the introduction in grammatically incorrect. “It” cannot be used when not immediately followed by a referent.

The usage of CO2 and carbon dioxide should be consistent in the paper.

The language in the introduction section is suggested to be thoroughly polished by a native.

This paper is suggested for major revision.

Reviewer #2: This article measures the embodied carbon emissions between China and other major countries or regions from the perspective of the value-added chain and uses factor decomposition methods to analyze the calculation results. Although there are many studies on embodied carbon emissions from the perspective of the value-added chain, this article focuses on the analysis of the carbon emissions correlation between China and other regions, which also has certain research significance. The research method of the whole article is appropriate, and the logical structure is clear. It is recommended to publish it after minor revisions. Before proceeding further, I have some small suggestions:

1. The literature review should be expanded. Such as Hongguang Liu et al. studied the carbon emissions embodied in value added chains in China (Journal of Cleaner Production, 2015,103: 362-370). Su and Ang (2017; Energy Economics 65, 137-147) firstly propose the aggregate embodied intensity (AEI) framework by defining the AEI indicator as the ratio of embodied energy (or emissions) to embodied value added using the I-O framework. Recently, the AEI analysis has been further extended to the transmission layer by Su et al. (2019; Energy Economics 83, 345-360). The AEI indicator at the higher level can be represented as a weighted sum of the AEI indicators at the lower level. There are already studies using the AEI indicators at the country level, such as China (Su and Ang, 2017; Su et al., 2019) and India (Zhu et al., 2018; Applied Energy 230, 1545-1556), and at the global level, such as Yang and Su (2019; Applied Energy 253, 113552) and Duan and Yan (2019; Energy Economics 83, 540-554) using the WIOD database. These studies also use the SDA technique to investigate the driving forces to the changes observed.

2. Some detailed description of the data source should be given, such as the industry and region division information.

3. It is best to add a discussion section to discuss the comparison between the results of this article and similar research results.

6. PLOS authors have the option to publish the peer review history of their article (what does this mean?). If published, this will include your full peer review and any attached files.

Reviewer #1: **Yes: **Guoqian Chen

Reviewer #2: No

---

## [Decision Letter · Decision Letter 1]

8 Oct 2021

Research on China’s Embodied Carbon Import and Export Trade from the Perspective of Value-Added Trade

PONE-D-21-16951R1

Dear Dr. Yue,

We’re pleased to inform you that your manuscript has been judged scientifically suitable for publication and will be formally accepted for publication once it meets all outstanding technical requirements.

Kind regards,

Taoyuan Wei

Academic Editor

PLOS ONE

Reviewers' comments:

Reviewer's Responses to Questions

**Comments to the Author**

1. If the authors have adequately addressed your comments raised in a previous round of review and you feel that this manuscript is now acceptable for publication, you may indicate that here to bypass the “Comments to the Author” section, enter your conflict of interest statement in the “Confidential to Editor” section, and submit your "Accept" recommendation.

Reviewer #2: All comments have been addressed

2. Is the manuscript technically sound, and do the data support the conclusions?

Reviewer #2: Yes

3. Has the statistical analysis been performed appropriately and rigorously? 

Reviewer #2: Yes

4. Have the authors made all data underlying the findings in their manuscript fully available?

Reviewer #2: Yes

5. Is the manuscript presented in an intelligible fashion and written in standard English?

Reviewer #2: Yes

6. Review Comments to the Author

Reviewer #2: The author has made changes based on the comments in the previous round, and I have no comments in this round.

7. PLOS authors have the option to publish the peer review history of their article (what does this mean?). If published, this will include your full peer review and any attached files.

Reviewer #2: No

---

## [Editor Report · Acceptance letter]

14 Oct 2021

PONE-D-21-16951R1 

Research on China’s Embodied Carbon Import and Export Trade from the Perspective of Value-Added Trade 

Dear Dr. Yue:

I'm pleased to inform you that your manuscript has been deemed suitable for publication in PLOS ONE. Congratulations! Your manuscript is now with our production department. 

Kind regards, 

on behalf of

Dr. Taoyuan Wei 

Academic Editor

PLOS ONE